# Magnesium-Rich Calcium Phosphate Derived from Tilapia Bone Has Superior Osteogenic Potential

**DOI:** 10.3390/jfb14070390

**Published:** 2023-07-24

**Authors:** Xiaxin Cao, Jiaqi Zhu, Changze Zhang, Jiaru Xian, Mengting Li, Swastina Nath Varma, Ziyu Qin, Qiaoyuan Deng, Xinyue Zhang, Wei Yang, Chaozong Liu

**Affiliations:** 1Hainan Provincial Fine Chemical Engineering Research Center, Hainan University, Haikou 570228, China; 20203105022@hainanu.edu.cn (X.C.); 22210710000051@hainanu.edu.cn (J.Z.); 20203100926@hainanu.edu.cn (C.Z.); 21210817000024@hainanu.edu.cn (J.X.); ziyuqin@hainanu.edu.cn (Z.Q.); xinyuezhang@hananu.edu.cn (X.Z.); yw@esfish.com (W.Y.); 2Institute of Orthopaedic & Musculoskeletal Science, University College London, Royal National Orthopaedic Hospital, London HA7 4LP, UK; t.varma@ucl.ac.uk; 3Key Laboratory of Advanced Material of Tropical Island Resources of Educational Ministry School of Materials Science and Engineering, Hainan University, Haikou 570228, China; qydeng@hainanu.edu.cn; 4Hainan Xiangtai Fishery Co., Ltd., South of Yutang Road, Industrial Avenue, Laocheng Development Zone, Chengmai City 571924, China

**Keywords:** tilapia bone, gradient thermal sintering, calcium phosphate bioceramics, osteogenesis

## Abstract

We extracted magnesium-rich calcium phosphate bioceramics from tilapia bone using a gradient thermal treatment approach and investigated their chemical and physicochemical properties. X-ray diffraction showed that tilapia fish bone-derived hydroxyapatite (FHA) was generated through the first stage of thermal processing at 600–800 °C. Using FHA as a precursor, fish bone biphasic calcium phosphate (FBCP) was produced after the second stage of thermal processing at 900–1200 °C. The beta-tricalcium phosphate content in the FBCP increased with an increasing calcination temperature. The fact that the lattice spacing of the FHA and FBCP was smaller than that of commercial hydroxyapatite (CHA) suggests that Mg-substituted calcium phosphate was produced via the gradient thermal treatment. Both the FHA and FBCP contained considerable quantities of magnesium, with the FHA having a higher concentration. In addition, the FHA and FBCP, particularly the FBCP, degraded faster than the CHA. After one day of degradation, both the FHA and FBCP released Mg^2+^, with cumulative amounts of 4.38 mg/L and 0.58 mg/L, respectively. Furthermore, the FHA and FBCP demonstrated superior bone-like apatite formation; they are non-toxic and exhibit better osteoconductive activity than the CHA. In light of our findings, bioceramics originating from tilapia bone appear to be promising in biomedical applications such as fabricating tissue engineering scaffolds.

## 1. Introduction

In recent years, bone fracture has become increasingly frequent. In 2019 alone, there were 178 million new cases of fracture all over the world, which is an increase of 33.4% compared with 1990 [1]. Thus, the materials used for bone repair are in great demand. Autografting is the gold standard of bone grafting [2,3]. However, some doubts surrounding autografting are associated with some complications, such as haematoma formation, blood loss, nerve injury, hernia formation, and chronic pain at the donor site [3]. Furthermore, in some circumstances, the autologous bone graft material may be deficient [4].

Owing to their excellent biocompatibility and bone conductivity, calcium phosphate ceramics such as hydroxyapatite (HA), biphasic calcium phosphate (BCP), and beta-tricalcium phosphate (β-TCP) have been used as bone repair materials [5,6,7]. Synthetic calcium phosphate is the main source of calcium phosphate owing to its easily controlled chemical composition and phase composition; however, its biological activity is potentially limited by its deficiency in trace elements and nanostructures compared to natural calcium phosphate materials [8,9]. To address these deficiencies, an increasing number of researchers have attempted to incorporate trace elements such as magnesium, iron, and strontium into hydroxyapatite [10,11,12]. Gong et al. synthesised Mg-doped hydroxyapatite whiskers which promote cell proliferation and differentiation [13]. Qi et al. developed strontium-substituted hydroxyapatite (SrHA) using a hydrothermal technique. The synthesised SrHA was then employed in the 3D fabrication of PLLA/SrHA composite scaffolds. The PLLA/SrHA scaffolds exhibited sustained Sr^2+^ release, with a cumulative concentration of 5.6 mg/L over 28 days which significantly enhanced the bioactivity of the PLLA/HA [14]. However, this form of doping cannot fully mimic the trace elements that exist in natural bone.

In the past, bovine bone was regarded as the optimal natural resource for producing natural calcium phosphate bioceramics. It possesses exceptional bone-repair properties and was used extensively in periodontal and implant surgery [15,16]. Due to mad cow disease (BSE), however, the safety of bovine-derived bone was questioned [17]. Since the 1970s, there has been study and application of the use of marine species as sources for biomaterials [18]. Early research studies indicate that corals [19], sponges [20], cuttlebone [21], and sea urchin spines [22] possess great potential as bone substitutes. Three important reasons why bioceramics derived from marine fish are being extensively researched for use as bone fillers are as follows: (1) they are generally composed of calcium carbonate in the form of aragonite or calcite, which are easily converted into calcium phosphates via hydrothermal conversion; (2) they have unique porous structures that are useful for cell migration and liquid penetration; and (3) they are rich sources of trace elements. According to Pujie Shi et al.’s studies, salmon, cod, and rainbow trout bones were heated to 650 °C for five hours to produce pure HA, which had significantly improved biological properties compared to other materials [9]. From tilapia bone, Modolon et al. successfully produced nanostructured biological hydroxyapatite with a variety of trace elements [23]. Through the calcination of various tilapia bones of varying ages, Weinand et al. were able to synthesise biphasic calcium phosphate (BCP) ceramics, which exhibit impressive alveolar bone regenerating capability and non-cytotoxicity [24].

In this study, fish bone hydroxyapatite (FHA) and biphasic calcium phosphate (FBCP) were synthesised from tilapia bone using a simple and novel gradient calcination approach. The physicochemical characteristics, including the morphology, components, degradation, and biomineralisation properties, of the tilapia bone-derived bioceramics were studied in depth. In vitro cytotoxicity and osteogenesis assessments were performed to evaluate biocompatibility and osteoinductive activity.

## 2. Materials and Methods

### 2.1. Materials

Tilapia bones were collected from Hainan Xiangtai Fishery Co., Ltd., located in Chengmai, Hainan Province, China. They were used as raw materials to produce bioceramics. Based on previous studies [25,26,27], the tilapia bones were boiled at 100 °C for 30 min to remove organic substances and the adherent fish meat. The bones were then cut into small pieces, dried in a hot-air oven, and prepared for use after drying.

Commercial hydroxyapatite (CHA, purity 98%, RHAWN), which was employed as a control, is a biomedically pure hydroxyapatite whisker.

### 2.2. Conversion of Tilapia Bone into Calcium Phosphate Bioceramics

The tilapia bones were placed in a muffle furnace for calcination and calcined at various temperatures (600 °C, 700 °C, and 800 °C) at a heating rate of 10 °C/min in air. Once the calcination temperature had been reached, the bones were maintained isothermally for 1 h to synthesise the FHA bioceramic. The tilapia bones were placed in a muffle furnace for calcination at various temperatures (600 °C, 700 °C, and 800 °C). To prepare the FHA bioceramics, the tilapia bones were sintered at the designated temperature for 1 h at a heating rate of 10 °C/min, then cooled to room temperature at 10 °C/min. Subsequently, the obtained FHA was used as a precursor and calcined at various temperatures (900 °C, 1000 °C, 1100 °C, and 1200 °C) for 4 h to develop FBCP bioceramics [28]. Finally, the obtained products were ground using an agate mortar and pestle, milled into a powder, and further characterized [29].

### 2.3. Characterizations

#### 2.3.1. X-ray Diffraction (XRD) Analysis

The phase compositions and crystal structures of the tilapia-derived bioceramics were analysed via X-ray diffraction (XRD, Rigaku Smart Lab, Japan) with Cu Kα1 radiation. The data were collected over an angular range from 10° to 80°, with a step size of 0.01° and a step time of 0.06 s at a voltage of 40 kV and a current of 30 mA. The phases were identified by comparing the experimental X-ray diffractions with the standards compiled by the International Centre for Diffraction Data (ICDD PDF No. 74-0566) and (ICDD PDF No. 09-0169) [30]. We calculated the relative amount of HA using the relative areas of three strong characteristic peaks. The formula is as follows [31]:$$\omega_{1}=\frac{I_{1}}{I}$$
where *I*_1_ is the reflection intensity of three characteristic peaks of HA or β-TCP, *I* is the sum of the reflection intensity of the three characteristic peaks of both HA and β-TCP, and *ω*_1_ is the relative amount of HA or β-TCP in the sample.

#### 2.3.2. Scanning Electron Microscopy (SEM) and Transmission Electron Microscopy (TEM)

The morphologies and chemical compositions of the tilapia-derived bioceramics and CHA were observed via the combination of scanning electron microscopy (SEM) (GeminiSEM 300, Zeiss, Oberkochen, Baden-Württemberg, Germany) and energy dispersive spectroscopy (EDS) [28].The samples were deposited onto conductive adhesive to observe their surface morphologies. The crystallography of various samples was investigated using a transmission electron microscope (TEM, JEM-2100F, JEOL, Akishima-shi, Tokyo Metropolis, Japan) [32], and their lattice spacings were calculated using a digital micrograph. The samples were ultrasonically dispersed in ethanol for 15 min before they were tested on the copper grid [33].

#### 2.3.3. Fourier Transform Infrared Spectroscopy (FTIR) Analysis

The functional groups of the samples were detected via Fourier transform infrared spectroscopy (FTIR) (TENSOR27, Bruker, Karlsruhe, Baden-Württemberg, Germany) in a wavenumber ranging from 400 cm^−1^ to 4000 cm^−1^ with a 4 cm^−1^ resolution [25]. The samples were mixed with KBr and pressed to carry out the FTIR test [9].

### 2.4. Degradation Test In Vitro

The in vitro degradation of the samples was carried out by measuring the weight loss of the samples after they were soaked in a PBS solution (phosphate-buffered solution). The CHA was used as a control [28]. Powder (30 mg) was added into the PBS solution (30 mL) in a centrifuge tube (1 mg/mL), and each centrifuge tube was then placed in a shaking water bath at 37 °C for 28 days. The in vitro degradation of the samples was tested by measuring the weight loss of the samples after soaking them in a PBS solution for 7, 14, 21, and 28 days. At each predetermined time, the samples were dried at 60 °C for 24 h, and the weight loss (%) was determined using the following formula: weight loss (%) = (w_0_ − w_1_)/w_0_ × 100%, where w_0_ and w_1_ represent the dry weight of the powder before and after immersion, respectively. Additionally, approximately 0.5 mL of the PBS solution was collected after 1 day and the concentrations of Ca^2+^ and Mg^2+^ were determined via inductively coupled plasma–optical emission spectroscopy (ICP-OES) (Thermo Fisher iCAP 7400) [34].

### 2.5. Mineralization Test In Vitro

The in vitro mineralization of the samples was assessed by immersing the samples (1 mg/mL) in an SBF solution (simulated body fluid) at 37 °C for 14 days, and the SBF solution was refreshed once every 3 days during the incubation period. After 14 days, the samples were dried at 60 °C for 24 h, and SEM was used to examine the growth of apatite on the surface. In addition, the samples’ crystal phase changes were analysed via XRD, and the differences in functional groups before and after mineralization were analysed using FTIR.

### 2.6. Cytotoxicity Test In Vitro

L929 cells were purchased from the Shanghai cell bank of the Chinese Academy of Sciences and used to evaluate the toxicity of the tilapia-derived bioceramics. L929 cells were maintained in Dulbecco’s modified eagle medium (DMEM), which containing 10% FBS and a 1% penicillin–streptomycin solution. The culture medium was incubated at 37 °C with 5% CO_2_-saturated humidity. All cells were cultured to reached 80% confluence for the subsequent experiments.

The CCK-8 (Dojindo, Kumamoto, Japan) assay was used to assess the cytotoxicity of the FHA and FBCP using the L929 cells [28]. In short, the cells were seeded into 24-well plates at a density of 1 × 10^4^ per well and incubated overnight at 37 °C in a 5% CO_2_ incubator. The samples with different concentrations (200 μg/mL, 400 μg/mL, 600 μg/mL, and 800 μg/mL) were added to the 24-well plate and incubated at 37 °C for another 24 h. After that, the culture medium was removed and washed 2–3 times with PBS. CCK-8 was added and cultured in the incubator at 37 °C for additional 2 h. Subsequently, the CCK-8 working solution was aspirated, and 400 μL of DMSO was added to each well in a 24-well plate and incubated on a shaker for 10 min. The absorbance value at 450 nm was detected using a microplate reader (Singapore, model number: Mutiskan Sky), and the cell viability was expressed as a percentage of the control group without treatment. Meanwhile, cell proliferation was further evaluated via live/dead staining. After being incubated with different samples at a concentration of 800 μg/mL for 24 h, the L929 cells were washed three times with PBS and stained for 10 min with acridine orange (AO; Sigma Aldrich, Louis Missouri, DE, USA) and propidium iodide (PI; Sigma Aldrich) solutions. Finally, the stained samples were observed under a fluorescence microscope (Leica DM50000B).

### 2.7. Osteogenic Differentiation Analyses

The osteogenic ability of the different samples (CHA, FHA, and FBCP) was measured using alkaline phosphate staining [9]. Before extraction, the samples were sterilized in 70% ethanol and placed under ultraviolet (UV) light for 30 min. The sterilized samples were immersed for 24 h in Dulbecco’s modified Eagle’s medium (DMEM, Thermo Fisher Scientific, Waltham, MA, USA) in order to obtain the leaching liquid, called the conditioned medium. BMSCs cells were seeded in a 6-well plate and cultured with a regular culture medium. After 12 h, the medium was replaced with extracts containing 10% FBS. The BMSCs cultured in the conventional medium were regarded as the control. The ALP produced from the cells in the presence of extracts of the CHA, FHA and FBCP was evaluated after 7 days of culture. The staining was measured with an ALP staining kit in accordance with the manufacturer’s instructions, and the stained cells were photographed under a microscope. The positive staining area was calculated using image J (V1.8.0.112) software.

### 2.8. Statistical Analysis

All obtained results were subjected to statistical analysis using Prism Software version 8 (GraphPad). All data are expressed as the means ± standard deviations. Comparisons between multiple groups were performed using a one-way analysis of variance (ANOVA) with Bonferroni’s multiple comparison test [28]. Values of *p* < 0.05 were considered statistically significant.

## 3. Results and Discussion

### 3.1. Analysis of the Morphologies and Compositions of FHA and FBCP

Tilapia-derived FHA and FBCP were prepared via a gradient thermal treatment technology. The initial phase of the conversion of tilapia bone into hydroxyapatite (HA) occurs between 600 °C and 800 °C, and the HA began to decompose into β-TCP when the calcination temperature reached 900 °C. Figure 1a shows the X-ray diffraction (XRD) patterns of the FHA samples prepared at various temperatures, and most of the diffraction peaks matched with the characteristic HA pattern (ICDD PDF No. 74-0566). With the increase in calcination temperature in the first stage, the diffraction peak of the FHA shows a narrow and sharp changing trend, suggesting that the crystallinity of the HA increased as the temperature rose. In the second stage, the FBCP was obtained by further calcining the FHA between 900 °C and 1200 °C, which caused the HA to partially decompose into β-TCP (ICDD PDF No. 09-0169) (Figure 1b). Furthermore, the typical peaks of FHA (2theta = 31.82°, 32.22°, and 32.96°) and FBCP (2theta = 27.91°, 31.13°, 31.81°, 32.20°, 32.95°, and 34.50°) are also displaced to higher angles, indicating that the lattice parameter was reduced, and lattice contraction occurred [35].

The amount of β-TCP in the FBCP tended to increase from 16.6% to 28.8% as the second gradient sintering temperature increased from 900 °C to 1200 °C. The relevant contents of β-TCP and HA are presented in Table 1. According to previous studies, β-TCP can be produced from the decomposition of HA when the calcination temperature is higher than 1100 °C [36,37]. However, in this study, β-TCP was generated when the second gradient sintering temperature reached 900 °C. This phenomenon may be attributed to the substitution of Ca^2+^ with Mg^2+^, which has a smaller ionic radius in FHA, resulting in crystal lattice distortion and thermal instability [36,38]. As a consequence, magnesium-rich HA could decompose into β-TCP at a lower calcination temperature.

The morphologies of tilapia bone-derived bioceramics and CHA were observed using a scanning electron microscope (SEM). As shown in Figure 2a, the SEM images of the CHA and FHA exhibit similar small granules. On the other hand, the FBCP grains are well joined together and exhibit a porous surface structure, indicating that during the second calcination process, small granules agglomerated and developed into larger ones with larger particle sizes. The corresponding EDS analysis revealed that both the FHA and FBCP contained Mg; however, the magnesium content of the FBCP (2.9 wt%) is lower than that of the FHA (12.9 wt%) because the second gradient of high-temperature calcination of the FBCP promoted the volatilisation of magnesium [39,40], thereby reducing the magnesium content of the FBCP.

The TEM results further confirmed the crystal phase of the magnesium-rich calcium phosphate bioceramics derived from tilapia bone. As shown in Figure 2b, the lattice spacing of the FHA is 0.279 nm, corresponding to the (2 1 1) plane, which is smaller than the (2 1 1) lattice spacing of the pure HA (0.28147 nm). As we know, BCP is composed of HA and β-TCP. In this study, the lattice spacings of 0.1822 nm and 0.2598 nm are ascribed to the (2 1 3) and (2 2 0) planes of HA and β-TCP, respectively, which are also smaller than the corresponding lattice spacings of pure HA (0.18403 nm) and pure β-TCP (0.2607 nm). This is due to the partial substitution of Ca^2+^ by Mg^2+^ in the FHA and FBCP, resulting in decreases in the d-spacings and lattice parameters [41,42]. This phenomenon is in accordance with the XRD results. Together with these findings, the gradient thermal treatment allowed for the production of magnesium-rich bioceramics derived from tilapia bones, with FHA having a greater magnesium content than FBCP.

### 3.2. In Vitro Study of the Degradation and Bioactivity of FHA and FBCP

The degradation behaviours of the CHA, FHA, and FBCP were evaluated by measuring their weight loss in PBS. We know that an ion exchange process in solution can result in the formation of a biologically active carbonate apatite layer. This indicates that during the process of degradation, sorption definitely occurred on the calcium phosphate bioceramics. As a result, the weight loss of the samples due to degradation is greater than the sorption of apatite layer from the solution. As shown in Figure 3a, the weight loss of the samples increased with an increase in the immersion time. After immersion for 28 days, the weight losses of the CHA, FHA, and FBCP were 3.5%, 6.91%, and 16.45%, respectively. These findings suggest that the FBCP presented the maximum mass loss. In addition, owing to the high magnesium concentration in the FHA, the solubility of the FHA is enhanced [43,44]. Therefore, the FHA presented a higher degradation rate than the CHA. Furthermore, as seen in Figure 3c, owing to the higher magnesium content in the FHA, the concentrations of Mg^2+^ release from the FHA (4.38 mg/L) were higher than those from the FBCP (0.58 mg/L) after 1 days. The bioinorganic magnesium cation (Mg^2+^), which is naturally present in bone tissues and plays a crucial role in numerous cellular functions, is essential for the regulation of protein synthesis, enzyme activation, and bone formation [45,46]. For instance, magnesium can considerably promote the development of human marrow stromal cells (MSCs) and the expression of endogenous bone morphogenetic protein [47,48,49]. Previous studies suggested that the addition of Mg to biomaterials could improve their dissolubility and promote osteogenic and angiogenic differentiation, making them more suitable for use as a bone substitute [50,51,52]. Furthermore, magnesium also plays a role in calcification, regulates the immune microenvironment, and alters pH when it comes to bone metabolism. Herein, we believe that the magnesium cation produced by the FHA and FBCP may act as a biochemical signal molecule to provide a superior microenvironment for bone tissue regeneration.

The bioactivity of FHA and FBCP was evaluated by investigating their phase compositions and surface morphologies after immersion in simulated body fluid (SBF) for 14 days (Figure 4a–d). The characteristic peaks of β-TCP in the FBCP were remarkably reduced when compared with the FBCP without SBF immersion (Figure 4a). On the other hand, the relative intensities of the characteristic peaks at 2θ = 31.76°, 32.18°, and 32.92° in the FHA and FBCP increased, which can be attributed to the HA diffracting planes (211), (112), and (300). FTIR was also employed to identify the functional groups present in the various samples. As shown in Figure 4b,c, compared with Fourier transform infrared spectroscopy (FTIR) results before and after SBF immersion, new absorption peaks at 874 cm^−1^, 1464 cm^−1^, and 1465 cm^−1^ emerged which corresponded to a typical CO_3_^2−^ vibration, suggesting that carbonate hydroxyapatite had precipitated on the samples [9,53,54]. The corresponding SEM images show that the surfaces of the FHA and FBCP became coarse and covered with a newly formed dense, plate-like pattern arranged in a regular manner to form the well-known cauliflower-shaped morphology. Therefore, the results confirmed that tilapia bone-derived bioceramics show superior apatite formation capacity.

### 3.3. Evaluation of Cell Proliferation and Differentiation of FHA and FBCP

The cytotoxicity of the FHA and FBCP was evaluated using L929 cells. The CCK-8 assay (Figure 5a) was used to quantitatively characterise cell proliferation capacity, and the findings showed that the cells were metabolically active. Both the FHA and FBCP specimens have demonstrated compatible viability to control group at concentrations of 200, 400, 600, and 800 µg/mL. It was observed that the cell viability percentage increased consistently for the FHA samples in all concentration groups. Additionally, the live/dead staining results (Figure 5b) agreed with the quantitative evaluation, demonstrating that both the FHA and FBCP had negligible cytotoxicity. The aforementioned findings suggest that bioceramics derived from tilapia bone have good biocompatibility and a wide range of application possibilities in bone tissue engineering.

ALP plays a critical role in early osteogenesis and hydrolyses various types of phosphates to promote cell maturation and calcification [55,56,57,58]. Therefore, in this study, the ALP activities of the BMSCs’ indirect co-cultures with samples were investigated to evaluate the osteoinductive properties of the bioceramics derived from tilapia bone. As shown in Figure 6a, although there were no significant differences in the levels of ALP activity between the FHA and FBCP groups on day 7, both were higher than that of the CHA group. Furthermore, in the FHA and FBCP groups, ALP staining was more intense than in the CHA group, which is in accordance with the ALP expression results. These results demonstrated that the FHA and FBCP were superior to the CHA in terms of osteogenesis. In general, the natural tilapia bone-derived FHA and FBCP facilitated cell proliferation and differentiation and the formation of mineralised tissue. These improved osteogenic properties may be attributed to the increased degradation of the FHA and FBCP, which resulted in a greater release of Ca^2+^. Our ICP-OES results (Figure 3b,c) revealed that the Mg^2+^ release concentration increased significantly from 0 to 4.38 mg/L on the first day. According to previous studies, a moderate Mg^2+^ concentration (0–100 mg/L) could improve bone regeneration, whereas free Mg^2+^ and a high Mg^2+^ content (above 400 mg/L) both inhibited cell adhesion, proliferation, and osteogenic differentiation [59,60]. Therefore, we believe that the Mg^2+^ released from the FHA could also facilitate the proliferation and differentiation of BMSCs. Further in vivo studies are required to confirm that magnesium ions released by the FHA can effectively stimulate vascularisation and bone regeneration.

Due to their porous structure, abundance of trace elements such as Mg^2+^, Sr^2+^, Na^+^, Cl^−^, and F^−^, and the ease of their conversion into calcium phosphate, these marine fish bones have shown great promise as bone substitutes. In recent years, marine fish bones have been extensively examined. For instance, the bones of the Japanese sea bream [61], Brazilian river fish [62], Atlantic swordfish [63], and Atlantic cod fish [64] were all successfully employed to obtain calcium phosphates. However, the sea water living conditions of marine fish will have a significant effect on the types and amounts of trace elements in fish bones, leading to variations in the compositions of bioceramics derived from marine fish [65]. Tilapia is one the most popular fish in China’s Hainan Province, and the Hainan Xiangtai company supplied tilapia fish bones for this investigation. The tilapia were farmed in net tanks under controlled environmental and feeding conditions. Therefore, the bone composition of the tilapia used in this study is relatively stable. In this study, magnesium-rich calcium phosphate bioceramics (FHA and FBCP) were extracted from the tilapia bone using a gradient thermal treatment approach. Although the gradient thermal treatment allowed for the production of magnesium-rich FHA and FBCP, the second gradient of high-temperature calcination may have resulted in the volatilisation of magnesium, thereby reducing the magnesium content of the FBCP. Therefore, the magnesium content of the FHA is substantially greater than that of the FBCP, and the release of Mg^2+^ from the FHA was greater than that of the FBCP. On the other hand, the FBCP is composed of β-TCP and HA, degrades faster and has a higher concentration of Ca^2+^ release than the FHA. According to our research, the FBCP and FHA were both more effective at promoting cell growth and the osteogenic differentiation of BMSCs than the commercial hydroxyapatite product. This indicates that tilapia bone has a great potential for being converted into highly valuable compounds that can be used as substitute materials for artificial bone.

## 4. Conclusions

In this study, natural magnesium-rich calcium phosphate bioceramics (FHA and FBCP) were extracted from tilapia bones using a thermal gradient treatment strategy. Together, these results confirm that Mg-substituted calcium phosphate was generated via this gradient thermal treatment and that the FHA contains more magnesium than the FBCP. Additionally, the FHA and FBCP have greater in vitro mineralisation and degradation capacities than CHA, which can result in a more favourable microenvironment for the proliferation and differentiation of BMSCs due to appropriate concentrations of Ca^2+^ and Mg^2+^. In conclusion, the magnesium-rich calcium phosphate bioceramics generated from tilapia are expected to have widespread application in bone repair and regeneration owing to their favourable mineralisation capability, acceptable degradation characteristics, continuous magnesium ion release, and good biological features.

## Figures and Tables

**Figure 1 jfb-14-00390-f001:**
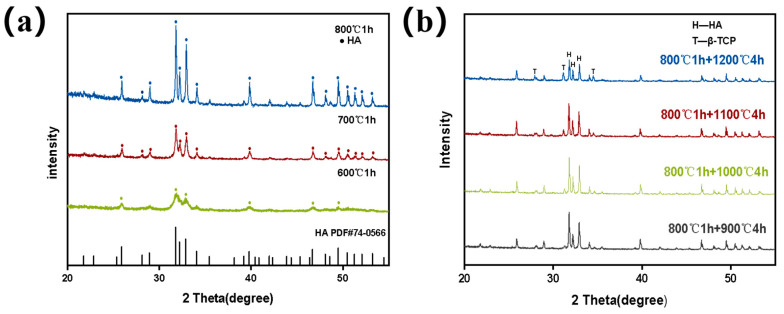
(**a**) XRD of FHA obtained via thermal processing at the first stage at various calcination temperatures (600 °C, 700 °C, and 800 °C). (**b**) XRD of FBCP obtained via thermal processing at the second stage at various calcination temperatures (900 °C, 1000 °C, 1100 °C, and 1200 °C).

**Figure 2 jfb-14-00390-f002:**
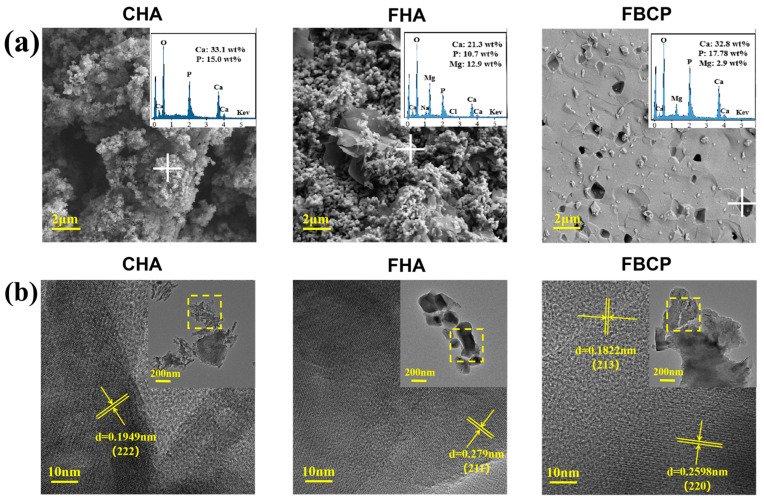
(**a**) SEM images of CHA, FHA, and FBCP and the corresponding EDS analyses (the white crosses show the EDS analysis points). (**b**) High-resolution TEM analyses of CHA, FHA, and FBCP (detected in the yellow box area of the inset TEM image).

**Figure 3 jfb-14-00390-f003:**
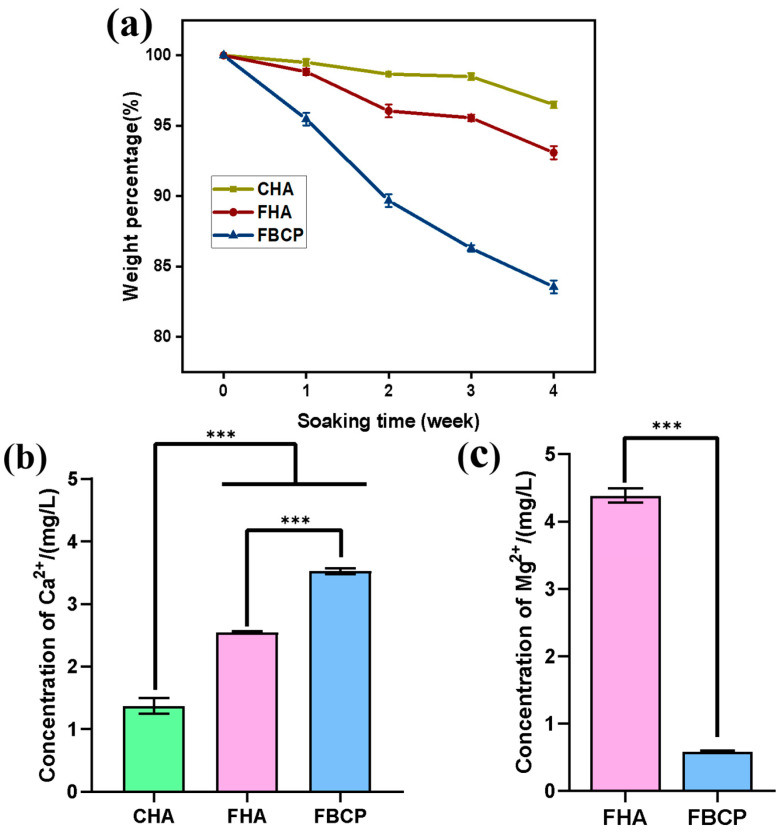
(**a**) The degradation behaviours of CHA, FHA, and FBCP in PBS. (**b**) Cumulative release of Ca^2+^ from CHA, FHA and FBCP after 1 day of degradation. (**c**) Cumulative release of Mg^2+^ from FHA and FBCP after 1 day of degradation. (*** *p* < 0.001).

**Figure 4 jfb-14-00390-f004:**
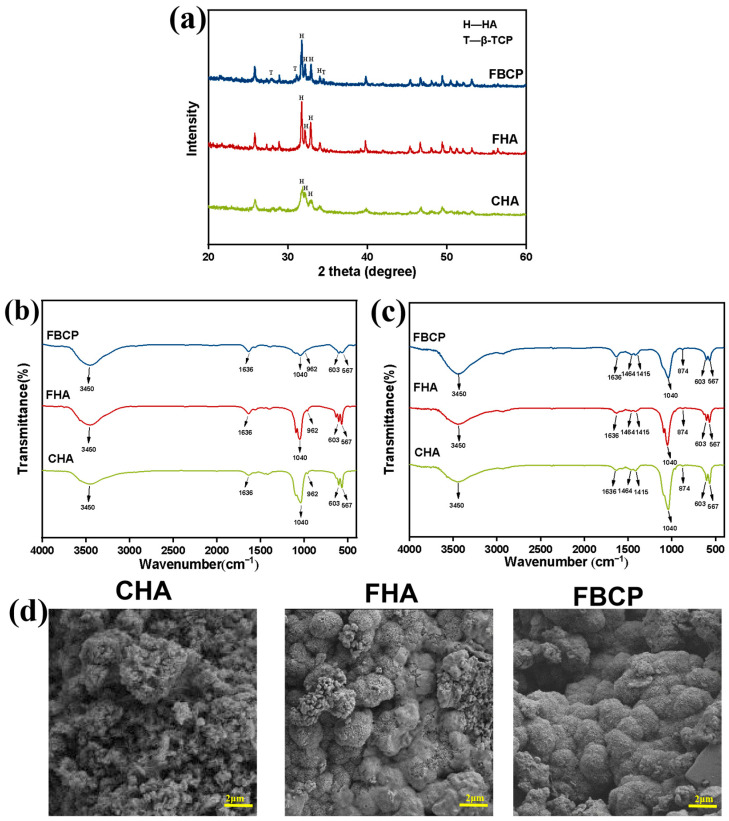
(**a**) XRD spectra after SBF immersion for 14 days. (**b**) FTIR spectrum before SBF immersion for 14 days. (**c**) FTIR spectrum after SBF immersion for 14 days. (**d**) SEM images after SBF immersion for 14 days.

**Figure 5 jfb-14-00390-f005:**
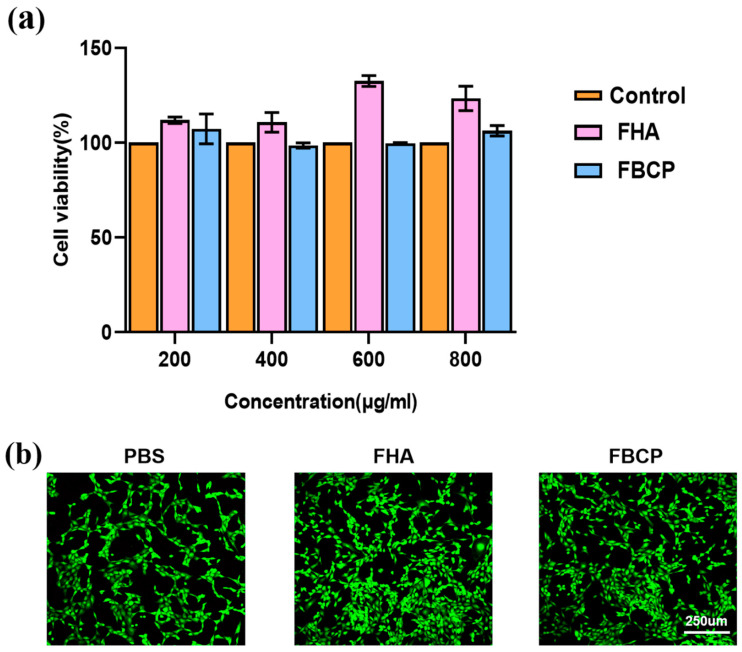
(**a**) Viability of cells with different concentrations of FHA and FBCP. (**b**) Live/dead staining of cells on FHA and FBCP.

**Figure 6 jfb-14-00390-f006:**
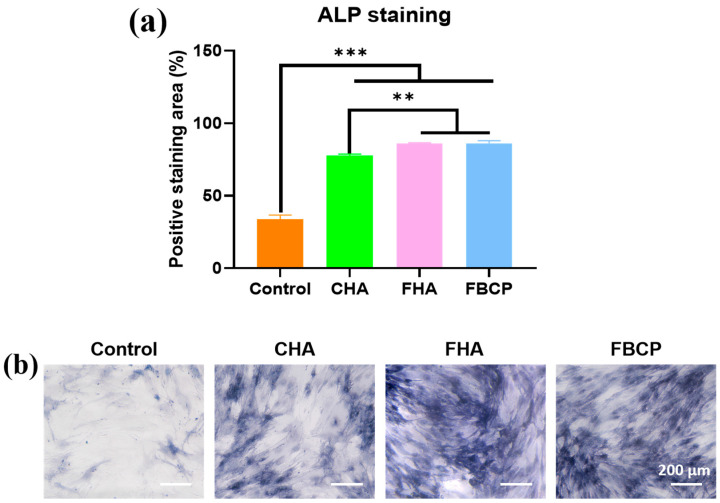
(**a**) ALP activity and (**b**) ALP staining images at 7 days of indirect cell culture with different samples (** *p* < 0.01 and *** *p* < 0.001).

**Table 1 jfb-14-00390-t001:** Relevant contents of FBCP and FHA.

Samples	Calcination Condition	β-TCP%	HA%
FHA	600 °C 1 h	0	100
	700 °C 1 h	0	100
	800 °C 1 h	0	100
FBCP	800 °C 1 h + 900 °C 4 h	16.6	83.4
	800 °C 1 h + 1000 °C 4 h	18.0	82.0
	800 °C 1 h + 1100 °C 4 h	20.7	79.3
	800 °C 1 h + 1200 °C 4 h	28.8	71.2

## Data Availability

The relevant data are already included in the article, and no additional data are required.

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
