# Peer review of "Magnesium-Rich Calcium Phosphate Derived from Tilapia Bone Has Superior Osteogenic Potential"

_jfb, 2023, doi:10.3390/jfb14070390_

Round 1

Reviewer 1 Report

Introduction.

 Several bone substitutes as calcium phosphate, hydroxyapatite (HA), biphasic calcium phosphate (BCP), and beta-tricalcium phosphate (β-TCP) have been used as bone repair materials by their excellent biocompatibility and osteoconductivity.

The authors report in this section the assessment of fish bone hydroxyapatite (FHA) and biphasic calcium phosphate (FBCP)as biomaterial for bone regeneration.  

 In this section the authors reported several studies with the incorrect names of the researchers:

Gong Liangzhi et al. synthetised Mg doped hydroxyapatite whiskers, which promote cell proliferation and differentiation [13]. Fangwei Qi developed strontium-substituted hydroxyapatite (SrHA) using a hydrothermal technique.

Modolon et al. successfully produced nanostructured biological hydroxyapatite with a variety of trace elements [23]. Through the calcination of various tilapia bones of varying ages, Wilson Ricardo Weinand et al. were able to synthesize biphasic calcium phosphate (BCP) ceramics, which exhibit impressive alveolar bone regenerating capability and non-cytotoxicity [24].

 The correct names are:

 Liangzhi et al [13]…..

 Qi et al [14]…..

Modolon et al [23]…..

 Weinand et al [24]…..

Materials and methods.

In the subsection 2.1.

The authors must report the full information about the tilapia bone  Hainan Xiangtai Fishery Co., Ltd, ( city, country)

The authors reported that  The tilapia bones were boiled at 100 °C for 30 84 minutes to remove organic substances. The bones were then cut into small pieces dried in 85 a hot air oven and ready for use after drying

The authors must report any related reference with this procedure. 

Similar informations related wirh each subsection of Material y methods (2.2.; 2.3, 2.4.; 2.5; 2.6.; 2.7) not showed any reference about each procedure. 

Results and Discussion. 

This section is very long and reports too figures.

 The authors showed a lot of results but a few and poor discussion. The authors must increase the discussion of the results with more international studies for to compare and to analyze the main experimental findings. There are several paragraphes without references of compared studies

 Conclusively, the study is not ready for publication.

Author Response

I would like to express our great gratitude to you and the referees, who have spent their valuable time to improve the quality of the manuscript. We would like to return the revised manuscript entitled “Magnesium-rich Calcium Phosphate Derived from Tilapia Bone Has Superior Osteogenic Potential” for consideration for publication in Journal of Functional Biomaterials.

The comments raised by the reviewers are very valuable and have been addressed carefully point-by-point in our response letter. We have also revised our manuscript with changes highlighted according to the comments from the reviewers.

Thank you again for your consideration of our manuscript for publication and look forward to hearing more news from you.

Reviewer 2 Report

Article “Magnesium-rich Calcium Phosphate Derived from Tilapia Bone Has Superior Osteogenic Potential” is based on a good idea and correctly conducted research. It is not written perfectly and there are some flows in discussion. But it is worth publishing after some improvements.

So here are few details that could be improved:

L33: Not universal but frequent.

L88: Not stove but furnace

L97: …a step size of 0.01, a scan speed of 10°…  If the step size is specified, then the time per step should be specified, not the scan speed. The scan speed is specified if the scan was continuous. Unit for scan speed is wrong anyway.

L99: JCPDS is the old name of ICDD. The name was changed in 1978. It should be …compiled by the International Centre for Diffraction Data (ICDD PDF No. 4-0566….

L100: Quantitative XRD data are reported later, method how quantitative data were obtained is not mentioned here.

L105: Not glued but deposited.

L105: Coper grid is used for TEM, not for SEM.

L10: …The crystallographic of … was investigated using TEM. Crystallography not crystallographic. Although mentioning of crystallography here is somewhat pretentious, only lattice spacings were determined.

L113: Avoid translucent, it is used only for visible light.

L184: L261: Why FHP and F-betaTCP, is F a surplus? Phases are reported here, not samples.

L199: Why gradient sintering temperature instead of just sintering temperature?

L203: “This phenomenon may be attributed to the substitution of a reduced ion radius Mg2+ for Ca2+ in FHA,…” I understand what you were trying to say, but it's a bit confusing. Maybe: …substitution of Ca2+ with Mg2+ having smaller ionic radius… Or something like that.

L207: Table 1 shows the same data as Figure 1a. One of that two is redundant.

L216: Is it justified to express the results of EDS analysis with two decimal places?

L217: Mg indeed has a slightly lower melting point and slightly higher vapor pressure. However, the theory of volatilization should be supported at least by literature references.

L219-228: XRD enables far more accurate calculations of lattice spacings and unit cell parameters. Why did the authors rely only on TEM?

L219-228: It is not always entirely clear to which crystal phase the statement refers.

L219-228: It's hard to tell, but if I understand correctly both the FHA sample, which is claimed to contain about 13% Mg, and the FBCP sample, which is claimed to contain about 3% Mg, show a significant reduction of lattice spacings in comparison to pure HA and beta-TCP. How do the authors explain this?

L234: …the weight losses of CHA, FHA, and FBCP were 3.5%, 6.91%, and 16.45%, respectively. These findings suggest that FBCP presents the maximum mass loss and has a higher concentration of Ca2+ release… I agree for mass loss, I don't see how it can be concluded from the data on mass loss that it loses more Ca. That claim could come after Fig 3c. What does even mean “higher concentration of Ca2+ release”?

L283: How come samples cause proliferation of mouse tissue cells. Plus, it seems that some confusion occurs due to a interchanging the concepts of proliferation and viability. Viability is a measure of the number of living cells in a population whereas proliferation is a measure of cell division. So, if I understand correctly, viability shouldn’t be so much greater than 100%. Small percentage is possible due to measurement error.

L305: Cell differentiation is the process of forming a variety of cell types that have specific functions. I don't see that being researched.

Kind regards

Here and there an unusual word appears, given the context.

Author Response

(The authors gave the same response as above.)

Reviewer 3 Report

Presented paper is devoted to the obtaining of HA from tilapia bone. The study seems interesting. However, there are number of comments to it.

Lines 87–92. Since authors presents the new method of HA and TCP synthesis, it is crucial to add more details in this section. For example, was heating occurred during 1 h with temperature increasing or after reaching 600/700/800 °C the samples were heat treated for 1 h?

Line 97. Is step size in degrees?

Line 103. Please ad information about CHA: manufacturer, purity, etc.

Line 115–129. Was sorption occurred on the samples?

Figure 1. Since there are no numerical values, please delete “a.u.” Same for Figure 4. Please add XRD pattern to the Figure 4c (like in Figure 4b).

Table 1. Please explain in more details, how content of samples was calculated.

Figure 2. Was EDS results obtained at point or by mapping?

Line 253. Is it appropriable to speak about bioactivity in this section? May be it will be better to change “bioactivity” to “biocompatibility”?

Figure 3. What is difference between (b) and (d)? These figures look similar. Moreover, cumulative release after day 1 and day 28 is the same. Please add data about Ca2+ release from CHA samples.

Author Response

(The authors gave the same response as above.)

Round 2

Reviewer 2 Report

The authors, more or less, accepted all my recommendations. I'm only concerned about the formula for the relative proportion of HA. I consider it a drastic oversimplification of the problem of quantitative XRD analysis. The reference 31 in no way suggests that the HA fraction could be calculated in this way. The areas under the XRD peaks are proportional to the amounts of the phases. However, this proportionality is not linear and that is why standards are used in quantitative XRD analysis, or the intensity ratio with the corundum peak given in ICDD data (Chung method) is used, or whole powder pattern decomposition analysis is performed. I absolutely do not agree with this method of calculation and the given formula.

Reviewer 3 Report

I sincerely thank the authors for providing accurate and detailed responses. There are only a few minor comments left.

Figure 2. Please indicate (for example, by cross) the point where EDS spectra were collected.

Please indicate in the paper that the weight loss of the samples as a result of degradation prevails over the sorption of HA from the solution.

I suppose that after correcting these remarks paper can be accepted.
